# Wilms Tumor: Updates about Pathogenesis and New Possible Clinical Treatments of the Most Frequent Pediatric Urogenital Cancer: A Narrative Review

**Giulio Perrotta [1,*]** and **Daniele Castellani [2]**

[1] Istituto per lo Studio delle Psicoterapie—ISP, 00185 Rome, Italy
[2] Urology Unit, Azienda Ospedaliero-Universitaria delle Marche, Università Politecnica delle Marche, 60126 Ancona, Italy; castellanidaniele@gmail.com
[*] Correspondence: info@giulioperrotta.com

**Abstract:** Background: Wilms tumor (or nephroblastoma) is a malignant and solid neoplasm that derives from the primitive renal bud. It represents the most frequent primary tumor of the urogenital tract in childhood, and treatment consists of surgery and chemo-radiotherapy. However, concerning quality of life, the new therapeutic frontier is exploring other safer and potentially more effective options, such as minimally invasive surgery and biological drugs. Method: Literature (PubMed) from January 2013 to July 2023 was reviewed, checking for innovations in diagnosis and treatment. Results: A total of 130 articles was included in the review. Conclusions: In addition to the therapeutic strategies already identified, such as classic surgery and pharmacological therapies, recent studies focus attention on the new frontiers of minimally invasive surgery, such as diagnostics using biomarkers and immunotherapy, which could represent a new therapeutic option and is possibly less risky than in the past, contributing in fact to the current knowledge of the scientific panorama in terms of "tumor microenvironment" and systemic implications deriving from oncological disease.

**Keywords:** Wilms; nephroblastoma; pediatric urogenital cancer; cancer

## 1. Introduction

### 1.1. General and Epidemiological Profiles

Wilms tumor (or nephroblastoma) is a solid malignant neoplasm that derives from the primitive renal bud. Wilms tumor represents the most frequent primary renal form of the urogenital tract in childhood [1] and can be unilateral (in 90–95%), bilateral or multicentric (in forms related to genetic factors), with both synchronous and metachronous presentation [2]. The prevalence of this tumor ranges between 2% and 6% among all childhood neoplasms [3], with an estimated worldwide prevalence of just over 1:10,000 children in Europe and North America and 4:10,000 in Asian countries [4,5], based on three characteristics that influence its epidemiological trend: (a) age (this tumor mainly affects children under the age of 15 with an average age at diagnosis between 2 and 5 years, and in general as many as 75% of cases occur before the age of 5); (b) gender (it is more frequent among females than males); (c) country of origin (it has a higher incidence in individuals of African origin, while it has a much lower incidence among Asians, with Europeans being in an intermediate position [6–8].

### 1.2. Pathology

From a macroscopic point of view, the tumor appears as a mass with well-defined margins, single or, more rarely, multiple, of a soft consistency; when cut, it appears grayish in color and homogeneous in appearance, although cysts, areas of necrosis or bleeding may be found inside. On the other hand, microscopic examination can reveal various aspects which recall the various stages of the embryological development of the kidney

(Figure 1). The most common picture consists of three cell types (epithelial, stromal and blastema) in various percentages. Each of these cell types can be present in different stages of differentiation: for example, epithelial cells can differentiate into tubular or glomerular cells, while stromal cells can remain undifferentiated or take on a fibrotic, myxoid or even skeletal muscle cell appearance (Figure 2) [9].

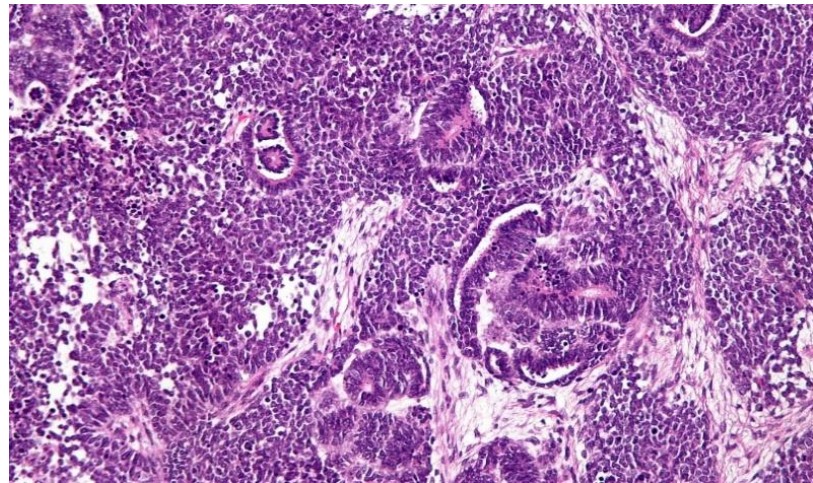

**Figure 1.** Classical histological features of WT include a triphasic pattern of epithelial, stromal and blastemal components. The image and description are from the AFIP Atlas of Tumor Pathology, according to entry #407018 in Pathology Education Instructional Resource. The Armed Forces Institute of Pathology Electronic Fascicles (CD-ROM Version of the Atlas of Tumor Pathology) contains U.S. Government work which may be used without restriction.

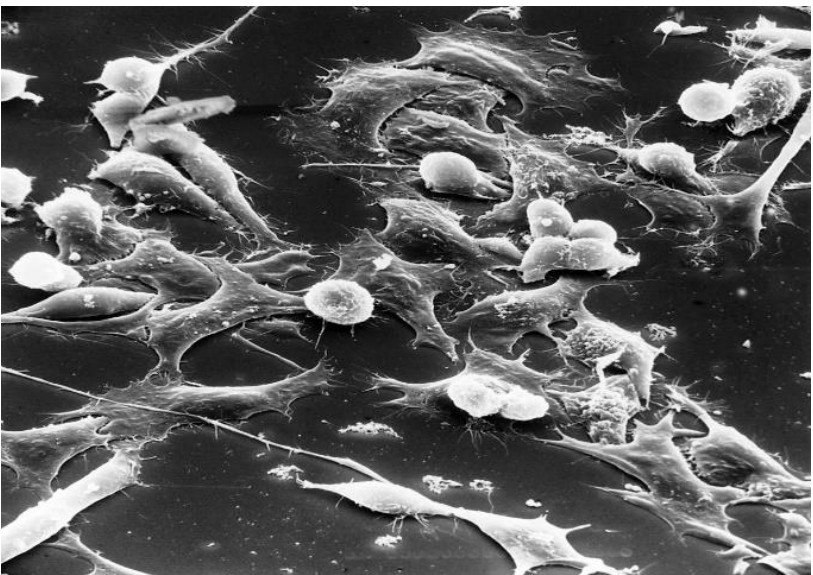

**Figure 2.** Wilms tumor. Scanning electron micrograph of cell culture about Wilms tumor, which is a human tumor that, like typical tumors, when attached to a surface is round. After it has been attached, it spreads out. The magnification is ×130. This image is in the public domain and can be freely reused. Source: https://visualsonline.cancer.gov/details.cfm?imageid=1763 (accessed on 7 September 2023).

### 1.3. Clinical Profiles

Unlike renal cell carcinoma, which is often asymptomatic, Wilms tumor often manifests itself in the form of a palpable abdominal mass, with a smooth surface and the presence

of abdominal pain. Systemic symptoms consist of fever, weight loss and fatigue, and less frequently also hematuria (due to internal bleeding) and hypertension (as a result of renin production by tumor cells or the activation of the renin–angiotensin–aldosterone axis due to reduced blood supply to the kidney when the tumor compresses the renal artery) can be present. Five stages of the disease are generally identified and defined based on the extent of the tumor and its eventual distant spread: [5,9–11].

Stage I: (a) The tumor is limited to the kidney. (b) The tumor is present in the perirenal fat but is surrounded by a fibrous (pseudo)capsule. The (pseudo)capsule might be infiltrated by a viable tumor which does not reach the outer surface. (c) The tumor might show protruding (botryoid) growth into the renal pelvis or the ureter but does not infiltrate their walls. (d) The vessels or the soft tissues of the renal sinus are not involved with the tumor. Intrarenal vessel involvement might be present.

Stage II: (a) A viable tumor is present in the perirenal fat and is not covered by a (pseudo)capsule, but it is completely resected (resection margins are clear). (b) The viable tumor infiltrates the soft tissues of the renal sinus. (c) The viable tumor infiltrates the blood and/or lymphatic vessels of the renal sinus or the perirenal tissue, but it is completely resected. (d) The viable tumor infiltrates the wall of the renal pelvis or the ureter. (e) The viable tumor infiltrates the vena cava or adjacent organs (except the adrenal gland) but is completely resected.

Stage III: (a) The viable tumor is present at a resection margin. A nonviable tumor or chemotherapy-induced changes present at a resection margin are not regarded as stage III. (b) Abdominal lymph node involvement is present by either viable or nonviable tumors. (c) Preoperative or intraoperative tumor rupture, if confirmed via microscopic examination (viable tumor at the surface of the specimen at the area of the rupture). (d) The viable or nonviable tumor thrombus is present at the resection margins of the ureter, renal vein or vena cava inferior (always discuss resection margins with the surgeon). (e) The viable or nonviable tumor thrombus, which is attached to the inferior vena cava wall, is removed piecemeal by a surgeon. (f) Wedge or open tumor biopsy before preoperative chemotherapy or surgery. (g) Tumor implants (viable or nonviable) are found anywhere in the abdomen. (h) The tumor (viable or nonviable) penetrates through the peritoneal surface.

Stage IV: Hematogenous metastases (for example, lung, liver, bone and brain) or lymph node metastases outside the abdominopelvic region.

Stage V: Bilateral renal tumors at diagnosis. Each side should be substage according to the above criteria.

Wilms tumor is often associated with other malformations or congenital defects of the urogenital system or other systems. It presents in Beckwith–Wiedemann syndrome, a genetic pathology usually caused by an imprinting defect (defect in the modulation of the expression of a part of the genetic material) or uniparental paternal disomy (when an individual receives two copies of a chromosome (or a whole chromosome and part of the second chromosome) from either parent and no chromosome copy from the other parent) for the 11p15 region, with an incidence of 1:15,000 births [12], or WAGR syndrome, which is characterized by mental retardation, aniridia and genital anomalies [13], or Denys–Drash syndrome, in which male pseudohermaphroditism is observed due to a defect in the development of the gonads and renal insufficiency [14].

## 2. Objective and Aim

This narrative review is intended to fulfil the primary research objective concerning the collection of the last decade's updates on the pathogenesis and treatments of this type of neoplasm, as it is the most frequent in the childhood age group and such findings are of clinical utility to present and future patients; secondary objectives relate to updates on pathogenesis and treatments. In particular, a review was conducted to determine the state of the art on this neoplasm and the implications derived from the discoveries regarding genetic predisposition and alternative therapies to classic surgery and chemo-radiotherapy, to verify innovations in diagnosis and treatment compared with the past.

## 3. Materials and Method

The authors searched PubMed, from January 2013 to July 2023, for meta-analyses, clinical trials and randomized controlled trials using the binomial "Wilms OR Nephroblastoma", selecting 190 eligibility results, to collect the updates of the last decade on pathogenesis and treatments about this type of neoplasm, to have a greater and complete overview of the topic, ultimately selecting a total of 79 studies, still adding 52 more reviews to be able to argue the elaborated content, for an overall total of 131 results. Simple reviews, opinion contributions or publications in popular volumes were excluded because they were not relevant or redundant for this work. The search was not limited to English-language papers (Figure 3).

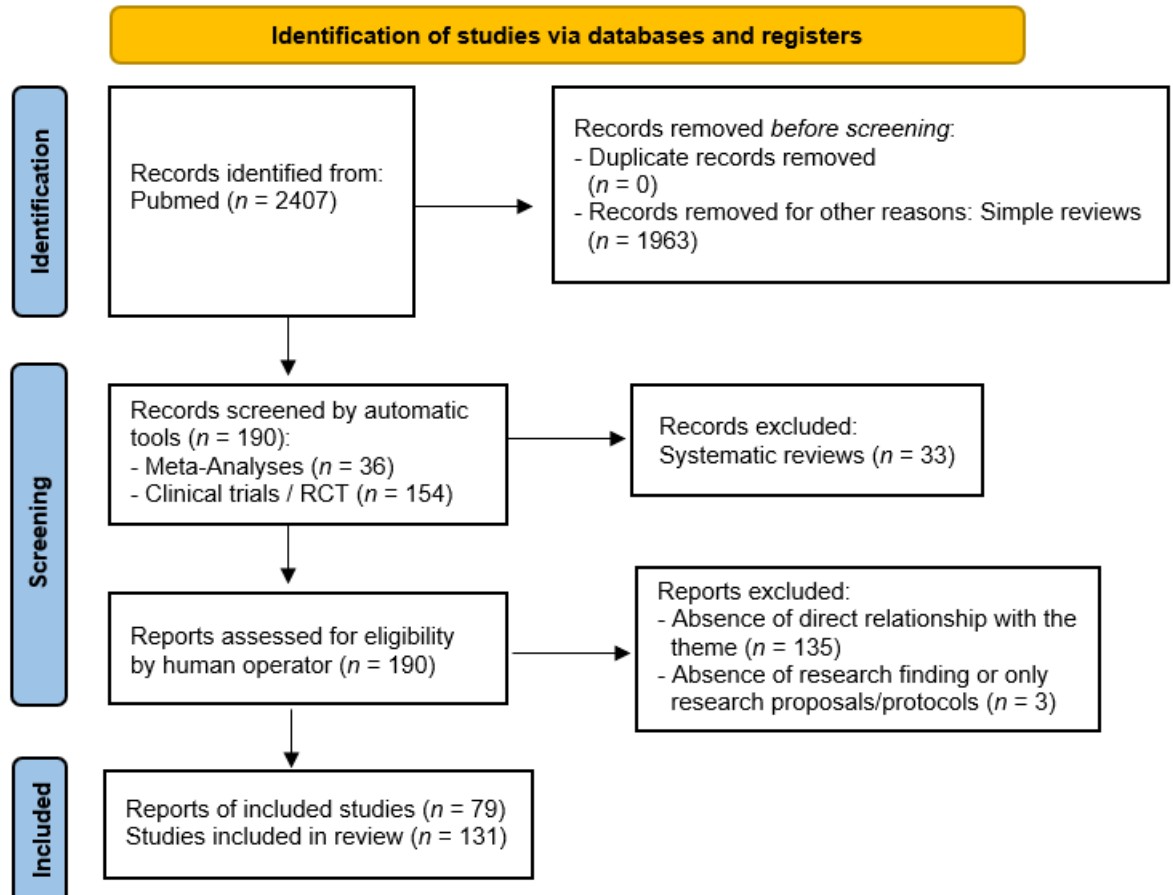

**Figure 3.** PRISMA flow diagram template for systematic reviews [15].

## 4. Results

### 4.1. Etiology

Most scientific literature agrees that Wilms tumor develops as a result of a genetic mutation. The implicated gene is called WT1 (from "Wilms Tumor") and is a tumor suppressor. The WT1 gene has been located on the short arm of chromosome 11 at position 11p13 and encodes a protein involved in the early stages of embryological development of the kidney (nephroblast differentiation). This gene is expressed only for a short period in embryogenesis, increasing during gestation, reaching a peak at birth and then decreasing significantly in the first 15 days after delivery (this regulation of gene expression is selective for the kidney) [16]. If mutated in both alleles, this gene causes tumor onset. In hereditary forms, the mutation on one of the alleles is inherited from one parent, while the mutation on the other allele is acquired later, while in sporadic (nonhereditary) forms both mutations are acquired [17]. Some gene mutations, on the other hand, lead to the production of a non-

functioning protein, thus determining a lack of control of cellular differentiation; however, when present in germ cells, these mutations cause severe developmental abnormalities of the kidneys and abnormalities of sexual differentiation, demonstrating the essential role of WT1 in the normal development of the urogenital tract. Other mutations are implicated in Wilms' tumor, such as the inactivation of the WTX gene located on the short arm of chromosome 11 at position 11p15, deletion of WT2 and loss of the heterozygosity of 16q and 1p [18–20], as well as the TP53 mutation being associated with a worse prognosis [21], while HMGA2 gene polymorphisms might weakly influence Wilms tumor predisposition, under certain circumstances [22]. Exposure to pesticides (such as organophosphates), vitamin deficiency in folic acid during pregnancy and maternal consumption of cigarettes and alcohol are finally related to this pathology [23,24]. As Wilms kidney tumors (WT) are genetically heterogeneous, tumors with WT1 mutations and concomitant loss of the wild-type allele represent yet another distinct subgroup frequently associated with mutations in CTNNB1. Gene expression profiling revealed that WT cell lines are highly human mesenchymal stem cells (MSCs), and FACS analysis demonstrated the expression of MSC-specific surface proteins CD105, CD90 and CD73. The stem cell-like nature of WT cells is further supported by their potential for adipogenic, chondrogenic, osteogenic and myogenic differentiation. Thus, it is concluded that WT with WT1 mutations exhibit specific features of PAM (multipotent mesenchymal precursors from paraxial mesoderm), which is the source of kidney stromal cells [25]. Recent research has identified a total of 636 high-quality small exonic variants as novel mutations in Wilms tumor; of these, 125 can be considered potentially harmful. The most frequently affected genes were those previously reported in WT in multiple studies: DROSHA, WT1, DGCR8, XPO5, AMER1, DICER1 and TRIM28, with mutations in MAX, MLLT1, MYCN, KRAS, SIX1 and SIX2 [26]. Alterations in DNA methylation, again, are molecular defects frequently observed in Wilms tumors, although it is unclear how these act in oncogenic function [27].

*4.2. Diagnosis*

Wilms tumor should always be suspected in the presence of a palpable abdominal mass in childhood, and abdominal ultrasonography is the simplest noninvasive diagnostic method that can aid in the differential diagnosis of hydronephrosis or other benign lesions of the kidney. On ultrasound, Wilms' tumor has the characteristics of a solid and inhomogeneous mass, in the context of which hypoechoic areas may be highlighted as the expression of intratumor necrosis [28]. On the other hand, contrast-enhanced computed tomography is more accurate and has the advantage of precisely defining the intra-abdominal extension of the tumor (for example, the invasion of lymph nodes or the presence of thrombi in the renal vein or the vena cava). Imaging plays an important role in the detection, staging, post-therapy evaluation and surveillance of Wilms tumor. Wilms tumor can be detected during the surveillance of a known cancer predisposition or after a child presents with symptoms. Magnetic resonance imaging defines the invasion of the great vessels much better than ultrasound [29] computed tomography [30] or Positron Emission Tomography [31] (although the choice of using the latter instrumental method is rarely considered in U.S. medical practice, except in cases of metastatic evidence beyond the second degree of spread). The use of biomarkers [32], including the prognostic biomarker circulating tumor DNA, seems promising but needs more insight into their use and validity [33–35]. The queen test (understood as the main and fundamental tool) for histological diagnosis, however, remains the biopsy, as to date it is the only diagnostic test that can define with certainty the nature of the biological finding and thus ensure the exact clinical approach to be followed. Histologic assignment to the different risk groups is based on the quantification of chemotherapy-induced changes, the percentages of the different viable components of Wilms tumor (epithelial, blastemal and stromal) and the presence or absence of anaplasia; here is the shared classification: low-risk (completely necrotic Wilms tumor); high-risk (blastemal type and diffuse anaplasia); intermediate-risk (all other types, by exclusion). In particular, in the intermediate-risk group, in addition to histological subclassification, the

tumor volume after preoperative chemotherapy (measured using imaging) is of importance for treatment stratification. If the tumor volume is >500 mL in stage II/III tumors of mixed type, regressive type or focal anaplasia type, these tumors are considered to be at higher risk of unfavorable outcomes and are treated aggressively to reduce the metastatic risk and subsequent surgical complications. Some 5% of Wilms tumors are multifocal (both unilateral and bilateral tumors), and they are more difficult to assess [36] (Table 1). Biopsy on the finding, however, in the case of the resectability of the tumor, can be performed after excision, failing the surgical requirement of the biopsy result, to avoid complications arising from incorrect procedure or time delays in scheduling surgery [36]. Equally important are nephrogenic remnants (NRs), which are persistent embryonic remnants in the kidney that are considered apparent precursors of Wilms tumor (WT). To date, two main categories of nephroblastomatosis (Nbl) have been recognized: perilobar (PLNR) and intralobar (ILNR). Hyperplastic NRs are much more common than previously thought and often confused with WTs, especially in cases of multicentric and bilateral tumors; in fact, biopsy itself is of limited value in distinguishing hyperplastic NRs from WTs, and even the use of surgery in Nbl cases requires careful clinical and surgical plan evaluation, as in many cases its role is questioned, reduced or cancelled by the use of chemotherapy [37].

**Table 1.** Histological criteria for Wilms tumor subtyping in International Society of Pediatric Oncology (SIOP) pretreated patients. The presence of diffuse anaplasia in any of the tumor types supersedes the subtypes; focal anaplasia also needs to be specifically mentioned.

| Tumor Type | Chemotherapy-Induced Change | Histological Features (% of Viable Tumor) | | |
|---|---|---|---|---|
| | | Blastema | Epithelium | Stroma |
| Completely necrotic | 100 | 0 | 0 | 0 |
| Regressive | >66 | 0–100 | 0–100 | 0–100 |
| Mixed | <66 | 0–65 | 0–65 | 0–65 |
| Mixed | <66 | 11–65 | 0–89 | 0–89 |
| Epithelial | <66 | 0–10 | 66–100 | 0–33 |
| Stromal | <66 | 0–10 | 0–33 | 66–100 |
| Blastemal | <66 | 66–100 | 0–33 | 0–33 |

Another characteristic feature of Wilms tumors is the marked tendency to invade blood vessels, in the form of a tumor thrombus, in the renal veins (with greater prevalence on the right, due to its smaller size than on the left), in the inferior vena cava and even in the right atrium, with 4–10% of cases along the renal vein in the inferior vena cava, while that in the atrium is relatively rare (under 1%). Diagnosis is made via imaging (computed axial tomography, magnetic resonance imaging and Doppler ultrasonography), as it is asymptomatic; in particular, ultrasonography is reliable for demonstrating the presence and extent of the tumor thrombus of the inferior vena cava. The management of Wilms tumor with tumor thrombus is determined using multiple factors such as the extent of the tumor thrombus and the chemotherapy response of the tumor, taking into account that the majority of intracavity and intra-atrial thrombi in Wilms tumor show a response to chemotherapy. Neoadjuvant chemotherapy causes tumor regression in almost half of patients, and most can be managed without the need for heart bypass surgery, while surgery is only indicated in a patient who is unstable due to thrombosis causing acute symptoms [38].

### 4.3. Therapy

Treatments are modulated according to clinical and anatomopathological variables, as well as according to national protocols specific to each national health system. Generally, the initial treatment of unilateral Wilms tumor is primary surgical resection (with an approach

that can be either anticipated or delayed based precisely on each patient's histologic and clinical outcomes) followed by adjuvant chemotherapy. A select group of younger patients with small tumors can be cured with surgery alone. The type of chemotherapy drug and the duration of therapy depend on the tumor histology and stage. Chemotherapy regimens depend on the risk group but usually consist of actinomycin D (dactinomycin) and vincristine, with or without doxorubicin, or adriamycin. For more aggressive tumors, intensive multi-agent chemotherapy regimens are used. Children with very large unresectable tumors or bilateral tumors are candidates for chemotherapy followed by re-evaluation and resection at a later time. Radiotherapy is given to children who have the disease at a higher stage (stage III and in the presence of distant metastases, usually of the lungs, which do not regress readily with chemotherapy). In most cases, radical nephrectomy is practiced, i.e., the surgical removal of the affected kidney, associated with the resection of the regional and para-aortic lymph nodes ipsilateral to the neoplasm; in bilateral tumors or patients with specific syndromes predisposed to the onset of nephroblastoma, partial nephrectomy is to be preferred whenever feasible [39–44]; when possible, especially in the case of bilaterality, preference should be given to even partial preservation of the renal structure, unless clinical conditions permit and the balance with the renal function to be preserved is compatible with possible tumor recurrence [45]. An improvement in the clinical picture and renal parenchyma over historical outcomes emerges in children with the bilateral form (but not in diffuse anaplasia and inhomogeneous tumors) if the treatment approach includes standardized three-drug preoperative chemotherapy, surgical resection within 12 weeks of diagnosis and response and postoperative therapy based on the histologic picture [46–51]; still, in the experimental phase, one study showed that concomitant administration of WT1-immunotherapy and standard neoadjuvant therapy (used in breast cancer) was well tolerated and induced WT1-specific antibodies in patients receiving aromatase inhibitors in the neoadjuvant phase (precisely for patients with Wilms tumor); however, in patients receiving neoadjuvant chemotherapy or the trastuzumab-chemotherapy combination, the humoral response was impaired or attenuated, probably due to the co-administration of corticosteroids and/or the chemotherapeutics themselves [52]. One study finally showed that radiofrequency with cryoablation was also effective for this tumor type [53]. To this day, however, immunotherapy and cryotherapy are not yet generally approved therapies in scientific communities such as the Children's Oncology Group and the Renal Tumor Study Group of the International Society of Pediatric Oncology (SIOP-RTSG), due to the few studies still in the literature, but it is hoped that in the future they may be considered as alternative and/or concomitant therapies to improve the quality of Wilms tumor treatments [54]. Another study has shown that the International Society of Pediatric Oncology (SIOP) protocol for Wilms tumor (WT), in the part that calls for preoperative chemotherapy as an initial approach, does not take into account the risk of histologic misdiagnosis and thus ineffective or even harmful treatment; therefore, it has been shown that fineneedle aspiration cytology (FNAC) can be a useful and feasible technique in children and can confirm the diagnostic suspicion of unilateral WT, avoiding inadequate preoperative chemotherapy in non-Wilms renal cancer [55]. Surgery, chemotherapy and radiotherapy essentially constitute the current treatment modality for Wilms tumor. The guidelines of the National Wilms Tumor Study Group (NWTSG/COG) and the International Society of Pediatric Oncology (SIOP) provide two different strategies for initial treatment, with some differences: in particular, in North America, the NWTSG/guidelines are adopted by the COG, while in Europe, it is usual to use the SIOP guidelines. Despite differences, determined from the individual case, the overall survival rate for patients treated using the two guidelines is similar, i.e., above 90% (in uncomplicated cases or with metastasis) [5]. Differences in approach between the different European and American currents, however, are resolved through the integration of practical experience, keeping in mind the specific characteristics of the patient's clinical history and the relevant guidelines. In the table, see the different treatments (Table 2):

**Table 2.** Differences between the COG and SIOP treatment protocols for WT. COG = National Wilms Tumor Study Group (NWTSG)/Children's Oncology Group; SIOP (The International Society of Pediatric Oncology) [5].

| Type | COG | SIOP |
|---|---|---|
| Surgery | Primary surgery before chemotherapy is recommended. For resectable tumors, preoperative or intraoperative biopsy is not performed, whereas in radical nephrectomy and lymph nodes, harvesting is performed through a transabdominal incision. To prevent tumor leakage, en bloc resection can be performed. The resection of a primary renal tumor should be considered even if at a stage IV disease (with metastases); renal-sparing surgery is not recommended, except in children with a solitary kidney, a predisposition to bilateral tumors or a horseshoe kidney, or in infants with Denys–Drash or Frasier syndrome (in order to delay the need for dialysis). | Radical nephrectomy of the tumor, performed after preoperative chemotherapy, is recommended. Lymph node sampling is important for staging, and the sampling of seven locoregional lymph nodes is necessary for accurate staging. Nephron-sparing surgery is used for nonsyndromic unilateral Wilms tumors, provided the following clinical conditions are met: (a) small tumor volume (<300 mL); (b) the expectation of substantial residual renal function in patients who have never had lymph node involvement. |
| Chemotherapy | Surgery is recommended as initial therapy before chemotherapy. Preoperative chemotherapy is indicated only in the following conditions: (a) with an inoperable Wilms tumor type; (b) with a solitary kidney; (c) with bilateral synchronous Wilms tumor; (d) a tumor thrombus in the inferior vena cava, extending above the level of the hepatic veins; (e) a tumor involving contiguous structures, whereby the removal of the kidney requires the removal of other organs (such as the spleen, pancreas or colon; (f) stage IV (with extensive pulmonary metastases). | Preoperative chemotherapy is recommended for all patients after diagnosis. For patients with unilateral localized tumors, a 4-week pretreatment is administered using vincristine (weekly) and dactinomycin (biweekly), while for patients with bilateral tumors, vincristine–dactinomycin is recommended for no more than 9 to 12 weeks (in some patients, doxorubicin is added as a reinforcer). Again, for patients with metastases, a regimen including 6 weeks of vincristine–#dactinomycin (as described above) and doxorubicin at weeks 1 and 5 is given. |
| Postoperative chemotherapy | It is recommended that postoperative chemo-therapy be used routinely in all patients with Wilms tumor except those at very low risk (those less than 2 years of age at diagnosis with a tumor, with favorable stage I histology, weighing less than 550 g, with sampling and with confirmed negative lymph nodes). | Postoperative chemotherapy is recommended in all patients with Wilms tumor except those with low-risk stage I tumors. |
| Postoperative radiation | Postoperative irradiation in the tumor bed is recommended for all patients with stage III cancer. | Radiation therapy of the whole abdomen is recommended for patients with intermediate-histology or high-risk tumors and with major tumor rupture preoperatively or intraoperatively, or with macroscopic peritoneal deposits. Lung radiotherapy is indicated for lung metastases without complete response until the 10th postoperative week. Patients with a complete response after induction chemotherapy with or without surgery do not need pulmonary radiotherapy. Patients with viable metastases at surgery or with high-risk histology require pulmonary radiation therapy. Whole-lung irradiation is recommended for patients who did not receive lung irradiation during first-line treatment, regardless of histology. |

**Table 2.** *Cont.*

| Type | COG | SIOP |
|---|---|---|
| Recurrent WT | Wilms tumors with characteristic high recurrence are divided into three risk groups: (1) standard risk; (2) high risk; (3) very high risk. In the first case (1), surgery (when possible), radiotherapy and chemotherapy (alternating cycles of vincristine/do-xorubicin/cyclophosphamide and etoposide/cyclophosphamide) are used. In the second and third cases (2,3), chemotherapy (alternating cycles of cyclophosphamide/ethoposide and carboplatin/ethoposide), surgery and/or radiotherapy and hematopoietic stem cell transplantation are recommended. | Patients with Wilms tumor are classified into AA, BB and CC, but essentially nothing changes from the previous classification. For the former (AA), only vincristine and/or dactinomycin is used as first-line treatment (without radiotherapy), with a four-drug regimen (combinations of doxorubicin and/or cyclophosphamide and carboplatin and/or etoposide); for the second group (BB), an intensive reinduction regimen (including the combination of etoposide and carboplatin with phosphamide or cyclophosphamide) is administered, followed by high-dose melphalan and autologous stem cell rescue or two more rounds of reinduction; for the third group (CC), camptothecins (irinotecan or topotecan) or new biologic compounds are recommended. |
| Stage V—WT | Both the COG and SIOP recommend preoperative chemotherapy and resection for bilateral WT. Bilateral renal-sparing surgery can be performed in patients with synchronous bilateral WT. Renal parenchyma sparing may help preserve renal function in these children. Renal transplantation is recommended and is usually delayed for 1–2 years without evidence of relapse. The SIOP also suggests that preoperative chemotherapy should be limited to no longer than 12 weeks, with time intervals for evaluation fixed to 6 weeks. | |
| Accepted Chemotherapy Regimens for Wilms Tumor | (1) Regimen EE-4A: Vincristine, dactinomycin × 18 weeks postnephrectomy [56]; <br> (2) Regimen DD-4A: Vincristine, dactinomycin, doxorubicin × 24 weeks; baseline nephrectomy or biopsy with subsequent nephrectomy [56]; <br> (3) Regimen I: Vincristine, doxorubicin, cyclophosphamide, etoposide × 24 weeks postnephrectomy [57]; <br> (4) Regimen M: Vincristine, dactinomycin, doxorubicin, cyclophosphamide and etoposide with subsequent radiation therapy [58]; <br> (5) Regimen UH1: Vincristine, doxorubicin, cyclophosphamide, carboplatin and etoposide × 30 weeks + radiation therapy [59]; <br> (6) Regimen UH2: Vincristine, doxorubicin, cyclophosphamide, carboplatin, etoposide, vincristine and irinotecan × 36 weeks + radiation therapy [59]. | |

The current research group, which replaces the National Wilms Tumor Study Group (NWRSG), is the Children's Oncology Group Renal Tumor Committee (COGRTC), adhering to the principles enucleated in the previous description [60].

The criteria for subclassifying pretreated WTs are as follows: The completely necrotic type shows no viable tumor elements. If more than 66% (two-thirds) of the tumor is non-viable (i.e., shows chemotherapy-induced changes), it is regarded as a regressive type, irrespective of the presence of remaining viable tumor components. If a viable tumor comprises more than one-third of the tumor mass, subtyping depends on the percentage of viable components: in mixed type, none of the components comprise more than 66% of the tumor; in epithelial (or stromal) type, in addition to having more than 66% of the tumor being composed of epithelial (or stromal) elements, the finding of only up to 10% of blastema is allowed (if the finding is more, then the tumor is subclassified as mixed type). Finally, preoperative chemotherapy is recommended for children with Wilms tumor with intravascular extension, as prolonged chemotherapy may improve resectability but increases tumor adhesion to the vascular endothelium, precluding complete resection. To evaluate the optimal duration of preoperative treatment, a study focused on overall survival (OS) and event-free survival (EFS), tumor regression, complete resection and cavectomy demonstrated that there is no evidence that cycle-prolonged periods of neoadjuvant chemotherapy confer additional benefits in treatment and are therefore not recommended (Table 3) [61].

Table 3. Current SIOP classification of pediatric renal tumors [61].

| RISK | Pretreated Tumors | Primary Nephrectomy Tumors |
|---|---|---|
| Low risk | Mesoblastic nephroma | Mesoblastic nephroma |
| | Cystic partially differentiated nephroblastoma | Cystic partially differentiated nephroblastoma |
| | Completely necrotic nephroblastoma | |
| Intermediate risk | Nephroblastoma—epithelial type | Nonanaplastic nephron-blastoma and its variants |
| | Nephroblastoma—stromal type | Nephroblastoma—focal anaplasia type |
| | Nephroblastoma—mixed type | |
| | Nephroblastoma—regressive type | |
| | Nephroblastoma—focal anaplasia type | |
| High risk | Nephroblastoma—blastemal type | Nephroblastoma—diffuse anaplasia type |
| | Nephroblastoma—diffuse anaplasia type | Clear cell sarcoma of the kidney |
| | Clear cell sarcoma of the kidney | Rhabdoid tumor of the kidney |
| | Rhabdoid tumor of the kidney | |

The classic triphasic Wilms tumor does not present diagnostic difficulties for pathologists, but when only one component is present (especially if small), the differential diagnosis may include renal cell carcinoma, metanephric adenoma and hyperplastic nephrogenic remnant, sarcoma of the kidney, mesoblastic nephroma and synovial sarcoma. Even pure Wilms tumor of the blastemal type may be difficult to distinguish from other embryonic "small round blue cell tumors", including neuroblastoma, primitive neuroectodermal tumor/Ewing's sarcoma, desmoplastic small round cell tumor and lymphoma. There are several reasons why making the correct diagnosis of Wilms tumor can be difficult: (1) the rarity of pediatric renal tumors results in a lack of experience with these entities for most pathologists; (2) the presence of different Wilms subtypes (morphological heterogeneity); (3) the morphological aspects can vary considerably from case to case; (4) the histological patterns of some Wilms subtypes may initially appear similar to those of other rare pediatric renal tumors; (5) the lack of clear-cut differential criteria distinguishing Wilms from nephrogenic remnants (NR), especially in limited biopsy material; (6) tumor evaluation and local pathological stage determination are multistep and time-consuming processes [62].

*4.4. Prognosis*

It is primarily linked to the histological aspect of the neoplasm, where the presence of anaplastic (undifferentiated) cells suggests a more unfavorable prognosis. The prognosis of Wilms tumor also depends on the stage at diagnosis and the patient's age (as older age is associated with a worse prognosis). Cure rates for low-stage (kidney-localized) disease range from 85 to 95%, and even children with more advanced diseases have a good prognosis: cure rates range from 60% (unfavorable histology) to 90% (favorable histology). However, the tumor can sometimes recur, generally within two years of diagnosis, although healing is also possible in children with relapsing cancer. The outcome after recurrence is better in children who present with a low-stage disease, whose tumors tend to recur in a site that has not been irradiated, who relapse more than one year after onset and who initially receive less intensive treatment [63]. Thus, the issue of tumor recurrence, especially in the hypothesis of bilaterality, is a negative index that warrants more careful and prolonged monitoring over time [64].

*4.5. Nutritional Implications*

The diagnosis of Wilms tumor implies a modification of the patient's nutritional plan, based on the symptoms and severity of the disease, such that to make a generalization is impossible, as each patient is a unique universe that requires a careful analysis of all factors

involved, starting from age and subjective anamnesis, following him in all stages of the disease process. A synergistic relationship will thus be established with any other therapies that may be prescribed, to promote the necessary functional integration, which could lead to a specific diet or a supplementary regimen of minerals and vitamins, including prebiotic and probiotic therapies capable of promoting the eubiosis of the organism which finds itself having to face cycles of decompensating pharmacological therapy [65,66].

*4.6. Psychological Implications*

As in all forms of oncology, the "cancer" event also necessarily impacts the psychological profile of the patient affected by the disease, just as it impacts the personal and relational life of his or her family members, considering the average age of the patient (of child range). Having to deal with the issue of death is always complicated, and for this reason, the family network must be supported at all stages of the disease, to ensure they have the minimum tools necessary to compensate for the distress caused by the disease. In the case of Wilms tumor, the final prognosis is good in most cases, especially if diagnosed in the early stages of the disease, and thus there is a greater chance of helping the patient and his or her family to overcome this complicated phase; however, the therapist must be prepared to deal with the possible distress that is grafted onto one or more personality frameworks of the people involved, perhaps already dysfunctional or decompensated by other pathologies, including mental pathologies, in psychophysical comorbidity. One needs to only think of neurotic pictures, characterized by anxious, phobic, somatic, obsessive or humoral (whether depressive or manic) outbursts, all the way to personality disorders. In more complex cases, irrespective of one's personality framework, the "cancer" event itself can trigger both acute distress (in its mildest forms) and adjustment disorder (following the diagnosis and processing process), all the way up to true post-traumatic stress, especially if the diagnosis is more unfavorable or not adequately managed by the clinical and family network. Therefore, the need to support the patient and family, from the earliest stages of the illness, is central to the healing process (including through cycles of psychotherapy and parental training techniques) and especially in the future perspective, and unfortunately, both in medical practice and in the literature, such profiles are often underestimated or relegated to mere services offered on par with organizational refreshments, effectively distorting the valuable clinical intervention on the cognitive and psychological sphere of the actors involved in the clinical stage [67–75].

In the extreme hypotheses of an inauspicious diagnosis, personal and family balances are upset, and one must take note of the new objective reality. Neoplastic disease is the most frequent cause of death and creates fear and anxiety in patients and relatives. The specificity of the tumor disease and the treatment undertaken for it are factors perceived and evaluated by the patient that influence his personality, defining specific moods and changes in his lifestyle. Cancer, more than any other disease, requires a continuous and repeated effort of adaptation on the part of the patient. Psychological adaptation is aimed at preserving one's mental and physical integrity, addressing reversible disorders and integrating irreversible ones. It consists of a series of cognitive, emotional and behavioral reactions [75]. In cancer patients, the efficacy of active listening, therapeutic interviewing, the use of psychometric instruments to calibrate certain measurements related to the subject's psychophysical well-being, targeted drug therapies and psychotherapeutic counselling techniques such as hypnosis, biofeedback, mindfulness and clinical approach schools have been demonstrated by decades of publications, with increasing results all encouraging, to support the difficult path of healing and acceptance of the disease [76–79].

## 5. Discussion and Prospects

The literature of the past decade, in addition to confirming the pathologic and clinical knowledge already known, has focused on the etiologic investigation of genetic biomarkers, as well as minimally invasive surgery and new biologic therapies. Indeed, several biomarkers of Wilms tumor have been confirmed and variations in prevalence have been

shown, but most of these studies were based on small (and therefore not representative) samples; in particular, they showed that the limited prevalence of currently known genetic alterations in Wilms tumors (especially in bilateral tumor hypotheses, with alterations in 11p15. 5, WT1, TRIM28 and REST, but also in LOH1p, 16q, 1q and LOH11p15) [80–82] indicate that significant factors of initiation and progression remain to be discovered, thus placing a marked emphasis on ethnicity as a source of heterogeneity. The literature is equally rich in contributions on the topic of classical surgery (because of the conservative need for renal organ structure concerning metastatic risk) and drug therapies (including chemotherapy in high-risk metastatic forms, radiotherapy in medium-low risk and chemo-radiotherapy in more aggressive and/or metastatic forms). As science has progressed, therefore, much attention has been paid to those qualitative profiles of the patient's life that were necessarily bent to the therapist's desire to promote a cure, depending on the metastatic risk this tumor has, as it is often asymptomatic to the negative impact on renal function. However, in patients with Wilms tumor, the high cure rate (>90%) achieved with the combination of surgery (including minimally invasive techniques [83,84] and ablation) and radio-chemotherapy has often been associated with high early and/or late toxicity and treatment-resistant clinical entities, such as disseminated anaplastic tumors or recurrent disease (including oncologic and systemic comorbidities), which still pose unresolved or difficult clinical problems due to the lack of specific research if not studies still in the experimental phase [44,45,85–89] or with specific protocols that nevertheless use small [90] or even insufficient population samples [91,92]. Another strand of study, which is extremely interesting, relates to the use of vaccinations specific for Wilms tumor peptide 1, which is involved in other cancer diseases (e.g., glioblastoma, pancreatic adenocarcinoma, promyelocytic leukemia, acute myeloid leukemia, certain types of head and neck cancer, malignant glioma, lung carcinoma, ovarian and uterine carcinoma and melanoma) [93–121] and myelodysplastic syndrome [122,123], just as it is in other morbid conditions with medium to high clinical risk. Still, another strand of study is then devoted to preoperative chemotherapy, with currently promising results [124], distinguishing between the three different tumor lineages (in fact, stromal and epithelial features have been shown to correlate with more favorable histology, while blastemal features are more unfavorable, as late-stage tumors shift precisely toward the renal blastemal archetype) [125,126]. The role of innovative telemedicine will also be crucial in the next decade, not only to experiment with new forms of care during hospitalization, but also to assist the patient during relocation to another facility or his or her home of residence, domicile or usual abode [127]. Finally, attention to psychological, nutritional and athletic profile [128,129] also becomes crucial for facilitating the patient's healing process, paying attention also to aspects often considered secondary, such as intestinal eubiosis [130] and dysbiotic processes that can facilitate the on-set of morbid conditions of an inflammatory and infectious nature, in a potentially complex clinical picture for the health of young patients. Further studies should be conducted on the usage of chemotherapy and radiotherapy under more accurate risk-stratified strategies to decrease the late effects of surgery [131].

In the table, see the clinical message, as the final result of the literature search (Table 4):

**Table 4.** Clinical message.

| Keyword | Clinical Message |
|---|---|
| Etiology | Most of the scientific literature agrees that Wilms tumor develops as a result of a genetic mutation (WT1, located on the short arm of chromosome 11 at position 11p13), and therefore a preventive genetic analysis could exclude the risk of being a carrier. Testing positive for the genetic test does not indicate certainty that the disease may occur during childhood, but correlations related to other genetic portions are also being investigated (e.g., 11p15. 5, WT1, TRIM28 and REST, but also LOH1p, 16q, 1q and LOH11p15); however, in the case of a positive result, attention should also be paid to other circumstances considered favorable and related to the disease: exposure to pesticides (such as organophosphates), folic acid deficiency during pregnancy and maternal consumption of cigarettes and alcohol or living in unhealthy environments. |

**Table 4.** *Cont.*

| Keyword | Clinical Message |
|---|---|
| Diagnosis | The first level of investigation is always the objective examination (with palpation) and abdominal ultrasound. In the second level of proceeding, computed tomography and contrast will be used, both for in situ evaluation and for the vascular component, but this is better investigated with MRI, as it better defines the invasion of the great vessels, although the latter is rarely used in U.S. medical practice. The use of biomarkers, including the prognostic biomarker circulating tumor DNA, appears to be promising but still needs more investigation about its use and validity. |
| Therapy | Treatments are modulated according to clinical and anatomopathologic variables, as well as according to national protocols specific to each national health system. Generally, the initial treatment of unilateral Wilms tumor is primary surgical resection (with an approach that can be anticipated or delayed precisely according to each patient's histologic and clinical outcomes) followed by adjuvant chemotherapy. The type of chemotherapy drug and duration of therapy depends on the histology and stage of the tumor. Chemotherapy regimens depend on the risk group but usually consist of actinomycin D (dactinomycin) and vincristine, with or without doxorubicin, or adriamycin. For more aggressive tumors, intensive multiagent chemotherapy regimens are used. Radiation therapy is given to children who have more advanced-stage disease (stage III and in the presence of distant metastases, usually lung, that do not easily regress with chemotherapy). In most cases, radical nephrectomy, i.e., the surgical removal of the affected kidney, combined with the resection of regional and para-aortic lymph nodes ipsilateral to the neoplasm, is performed; in bilateral tumors or patients with specific syndromes predisposing them to nephroblastoma, partial nephrectomy is preferred when possible; when possible, especially in cases of bilaterality, even partial preservation of the renal structure should be preferred, unless the clinical conditions do not allow it and the balance with the renal function to be preserved is compatible with possible tumor recurrence. One study finally showed that radiofrequency with cryoablation is also effective for this type of tumor. However, immunotherapy and cryotherapy are not yet generally approved therapies in scientific communities, such as the Children's Oncology Group and the Renal Tumor Study Group of the International Society of Pediatric Oncology (SIOP-RTSG), because of the few studies still in the literature. |
| Prognosis | Wilms tumor is the most common renal tumor in childhood, and the prognosis is mainly related to the histologic appearance of the neoplasm, where the presence of anaplastic (undifferentiated) cells suggests a poorer prognosis. The prognosis of Wilms tumor also depends on the stage at diagnosis and the patient's age (advanced age is associated with a worse prognosis). Cure rates for low-stage disease (localized to the kidneys) range from 85 to 95 percent; children with the more advanced disease also have a good prognosis: cure rates range from 60 percent (unfavorable histology) to 90 percent (favorable histology). The tumor can sometimes recur, usually within two years of diagnosis, although a cure is possible even in children with recurrent tumors; the problem of tumor recurrence, especially in the case of bilaterality, is a negative index that warrants closer and more prolonged monitoring over time. |
| Nutritional implications | The diagnosis of Wilms tumor implies a modification of the patient's nutritional plan, depending on the symptoms and severity of the disease, such that generalization is impossible because each patient is a unique universe that requires careful analysis of all factors involved, starting with age and subjective history, following him or her through all stages of the disease process. |
| Psychological implications | The "cancer" event necessarily impacts the psychological profile of the patient affected by the disease, as well as the personal and relational life of his or her family members, also considering the patient's average age (of childhood range). In the case of Wilms tumor, the final prognosis is good in most cases, especially if diagnosed in the early stages of the disease, and thus there is a greater chance of helping the patient and his family to overcome this complicated phase; however, the therapist must be prepared to handle the possible distress that is grafted onto one or more personality frameworks of the people involved, perhaps already dysfunctional or decompensated by other pathologies, including mental pathologies, in psychophysical comorbidity. Therefore, the need to support the patient and family, from the earliest stages of the illness, is central to the healing process (including through cycles of psychotherapy and parent training techniques), even and especially in the future perspective. |

## 6. Conclusions

Minimally invasive surgery, as well as diagnostics using biomarkers and combined therapies, could represent a new and possibly less risky therapeutic option than in the past, effectively contributing to the current knowledge of the scientific landscape in terms

of "tumor microenvironment" and the resulting systemic implications from oncological disease. The research, in this regard, is still incomplete and immature and therefore requires further study and specific studies with significantly larger and more representative population samples, although these profiles nevertheless appear promising, especially because of the new single and combination therapies. On the other hand, there appears to be insufficient attention to therapeutic profiles aimed at restoring the eubiosis and psychological well-being of the patient and his or her family members, profiles that are almost always underestimated and in the background among the possible therapies suggested by health professionals.

**Author Contributions:** Conceptualization, G.P.; methodology, G.P. and D.C.; software, not applicable; validation, not applicable; formal analysis, G.P. and D.C.; investigation, G.P. and D.C.; resources, G.P.; data curation, G.P. and D.C.; writing—original draft preparation, G.P.; writing—review and editing, G.P.; visualization, G.P. and D.C.; supervision, D.C.; project administration, G.P. and D.C.; funding acquisition, not applicable. All authors have read and agreed to the published version of the manuscript.

**Funding:** This research received no external funding.

**Institutional Review Board Statement:** Not applicable.

**Informed Consent Statement:** Not applicable.

**Data Availability Statement:** Not applicable.

**Conflicts of Interest:** The authors declare no conflict of interest.

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
