# Peer review of "Wilms Tumor: Updates about Pathogenesis and New Possible Clinical Treatments of the Most Frequent Pediatric Urogenital Cancer: A Narrative Review"

_2673-4095, doi:10.3390/surgeries4040064_

Round 1
Reviewer 1 Report
Comments and Suggestions for Authors
The review manuscript attempts to cover all aspects of Wilms's tumor.
As the focus of the paper, the authors may need to convince the readers, by showing evidence, on how effective is active listening, therapeutic talk, the use of standardized psychometric tools, targeted pharmacological therapies and psychotherapeutic counselling, mindfulness, biofeedback and hypnosis techniques are in improving patients' wellness.
Instead of trying to be encyclopedic, the conclusion paragraph would be more effective if the readers take home one or two important messages.
Author Response
Dear Reviewer, I thank you for your suggestions. We will revise the text with this line in mind, sending the manuscript with corrections, hoping it will satisfy you.
Reviewer 2 Report
Comments and Suggestions for Authors
1. Etiology is over simplified and the whole relationship to precursor lesions is not mentioned. No mention of the multiple genetic aberration and methylation. No mention of the different genetic aberrations with anaplastic Wilms.
2. Pg 2 line 46- blastema
3. Objective to outline their primary focus
4. Line 171 refers to “queen test”- needs to be defined. The whole issue of biopsy is not clear- in that biopsy is not done in North America since if the tumor is resectable this will be done at the time of diagnosis whereas in Europe is not clear if a biopsy is needed prior to starting treatment.
5. Line 174 refers to the European approach and not to the North American approach; risk category is defined according to the European approach.
6. Line 192 “3”- not clear what this refers to.
7. Lines 225-231 refer to WT-immunotherapy for breast cancer reference #42- it is not relevant to this discussion and actually standard therapy has not even been presented.
8. Lines 231+ refer to cryotherapy and immunotherapy that are not part of treatment for Wilms tumor and not relevant to this discussion;
9. Standard therapy is not clearly discussed with outcomes (and particularly before discussing anything experimental.
10. Line 243- National Wilms Tumor Study Group – is in the past- the current research group is the Childrens Oncology Group Renal Tumor Committee;
11. Discussion and contrast between North American approach and European approach does not occur until lines 232—242; should be much earlier
12. Table 2 is comparing and contrasting approaches but is missing details such as approach to metastatic disease; radiation for stage 3- flank vs whole abdomen and how spill plays a role;
13. No references to treatment regimens in footnotes for both North American or Europe. For example if the reader wanted to understand more about regimen M-no reference is provided. No EFS/OS with references- only very broad outcomes;
14. Much more detail is needed for the treatment of tumor thrombus- much wider experience than what is reflected here.
15. Pages 7 and 8 no reference provided in table for treatment and outcomes
16. Pg 8 nephrogenic rests are mentioned but defines and significance is not detailed
17. Pg 9 reference to molecular genetic signatures but not mention of the significance of LOH 1 p, 16 q or 1q gain or LOH11p15 in very low risk Wilms tumor; Prognostic factors are not described; significance of some histologic subtypes;
Author Response
Dear Reviewer, I thank you for your suggestions. We will revise the text with this line in mind, sending the manuscript with corrections, hoping it will satisfy you. Thank you especially for the timeliness of your suggestions and for going into detail so that we can improve our work and offer readers a document worthy of attention. Cordial greetings
Round 2
Reviewer 2 Report
Comments and Suggestions for Authors
Did the authors provide a point by point response to the comments?
I did not see one in this packet.
Author Response
Dear, as requested I proceed point by point to the answers (which you will find in bold and highlighted to distinguish them more easily from previous requests):
- Etiology is over simplified and the whole relationship to precursor lesions is not mentioned. No mention of the multiple genetic aberration and methylation. No mention of the different genetic aberrations with anaplastic Wilms. Thank you for this clarification. I have expanded the section, inserting 2 new references, just on the missing points (lines 163-170, notes 25 and 26)
- Pg 2 line 46- blastema. I corrected, as indicated (lines 45-46)
- Objective to outline their primary focus Thank you for this clarification. I have expanded the section (lines 109-113)
- Line 171 refers to “queen test”- needs to be defined. The whole issue of biopsy is not clear- in that biopsy is not done in North America since if the tumor is resectable this will be done at the time of diagnosis whereas in Europe is not clear if a biopsy is needed prior to starting treatment. Grazie per la specifica. The section has been revamped and expanded in order to meet the need for revision (lines 189-215).
- Line 174 refers to the European approach and not to the North American approach; risk category is defined according to the European approach. This part was also clarified and modified (lines 183-188).
- Line 192 “3”- not clear what this refers to. It was a typo and has been removed.
- Lines 225-231 refer to WT-immunotherapy for breast cancer reference #42- it is not relevant to this discussion and actually standard therapy has not even been presented. I agree with the reviewer and the section has been modified to meet the review requirements by removing the specification.
- Lines 231+ refer to cryotherapy and immunotherapy that are not part of treatment for Wilms tumor and not relevant to this discussion; I agree with the reviewer and the section has been modified to meet the review requirements by removing the specification.
- Standard therapy is not clearly discussed with outcomes (and particularly before discussing anything experimental.
The present narrative work does not delve into all the issues related to this depth of detail, otherwise a systematic or meta-analysis would have been chosen. Simply, the data already reported are in our opinion sufficient to give the reader a detailed and illustrative overview of Wilms' tumor, and further additions would make the reading heavier by having to modify the structure of the manuscript. The point raised by the reviewer is nevertheless addressed in several passages of the manuscript, adding even the issues he raised about the vascular component.
- Line 243- National Wilms Tumor Study Group – is in the past- the current research group is the Childrens Oncology Group Renal Tumor Committee; Required changes were made, updating the manuscript (lines 281-298 + tab 2)
- Discussion and contrast between North American approach and European approach does not occur until lines 232—242; should be much earlier. This has been clarified and changed throughout the manuscript, not just in the lines identified by the reviewer, so as to clarify in detail the differences in application.
- Table 2 is comparing and contrasting approaches but is missing details such as approach to metastatic disease; radiation for stage 3- flank vs whole abdomen and how spill plays a role; The table has been edited and updated in every detail to eliminate contrasts and better explain to the reader the details in case of metastasis and staging levels.
- No references to treatment regimens in footnotes for both North American or Europe. For example if the reader wanted to understand more about regimen M-no reference is provided. No EFS/OS with references- only very broad outcomes; Several references have been added, changing the entire text, including the abstract and the materials and methods and references section (since there are dozens of changed parts in the text you will find all changes highlighted in yellow).
- Much more detail is needed for the treatment of tumor thrombus- much wider experience than what is reflected here. A section entirely devoted to this has been added (lines 221-234 and several references throughout the text, including tables)
- Pages 7 and 8 no reference provided in table for treatment and outcomes. Throughout the text there are references and no section is uncovered. There is also the clinical message part of Table 4 that summarizes the contents in reference.
- Pg 8 nephrogenic rests are mentioned but defines and significance is not detailed. Two parts on this issue were specially included so as to respond to the reviewer and complete the manuscript (lines 207-215 + tab 2)
- Pg 9 reference to molecular genetic signatures but not mention of the significance of LOH 1 p, 16 q or 1q gain or LOH11p15 in very low risk Wilms tumor; Prognostic factors are not described; significance of some histologic subtypes; The manuscript contains all references on this topic, and the text has been edited to improve comprehension and timeliness (lines 163-170 + 406-407 + tab. 4).
With these changes I hope that the auditor may be satisfied, remaining at your complete disposal. Cordial greetings

Round 3
Reviewer 2 Report
Comments and Suggestions for Authors
none
Author Response
No comments.